# CircZXDC Promotes Vascular Smooth Muscle Cell Transdifferentiation via Regulating miRNA-125a-3p/ABCC6 in Moyamoya Disease

**DOI:** 10.3390/cells11233792

**Published:** 2022-11-26

**Authors:** Yuan Liu, Yimin Huang, Xincheng Zhang, Xiaopeng Ma, Xuejun He, Chao Gan, Xin Zou, Sheng Wang, Kai Shu, Ting Lei, Huaqiu Zhang

**Affiliations:** 1Department of Neurosurgery, Tongji Hospital, Tongji Medical College, Huazhong University of Science and Technology, Wuhan 430030, China; 2Institute of Integrated Traditional Chinese and Western Medicine, Tongji Hospital, Tongji Medical College, Huazhong University of Science and Technology, Wuhan 430030, China

**Keywords:** MMD, stroke, circRNA, ABCC6, ERS, transdifferentiation

## Abstract

Moyamoya disease (MMD) is an occlusive, chronic cerebrovascular disease affected by genetic mutation and the immune response. Furthermore, vascular smooth muscle cells (VSMCs) and endothelial cells (ECs) participate in the neointima of MMD, but the etiology and pathophysiological changes in MMD vessels remain largely unknown. Therefore, we established the circZXDC (ZXD family zinc finger C)–miR-125a-3p–ABCC6 (ATP-binding cassette subfamily C member 6) axis from public datasets and online tools based on “sponge-like” interaction mechanisms to investigate its possible role in VSMCs. The results from a series of in vitro experiments, such as dual luciferase reporter assays, cell transfection, CCK-8 assays, Transwell assays, and Western blotting, indicate a higher level of circZXDC in the MMD plasma, especially in those MMD patients with the RNF213 mutation. Moreover, circZXDC overexpression results in a VSMC phenotype switching toward a synthetic status, with increased proliferation and migration activity. CircZXDC sponges miR-125a-3p to increase ABCC6 expression, which induces ERS (endoplasmic reticulum stress), and subsequently regulates VSMC transdifferentiation from the contractive phenotype to the synthetic phenotype, contributing to the intima thickness of MMD vessels. Our findings provide insight into the pathophysiological mechanisms of MMD and indicate that the circZXDC–miR-125a-3p–ABCC6 axis plays a pivotal role in the progression of MMD. Furthermore, circZXDC might be a diagnostic biomarker and an ABCC6-specific inhibitor and has the potential to become a promising therapeutic option for MMD.

## 1. Introduction

Moyamoya disease (MMD) is a type of cerebrovascular disease named for its radiography feature of ‘smoke-like’ neovascularization, which is often associated with recurrent stroke. The major characteristic of MMD has been identified as the occlusion or stenosis of the internal carotid (ICA) at the entrance of the Willis circle [1]. According to several large clinical investigations, MMD is mainly prevalent in East Asia [2]. Current knowledge regarding the pathophysiological nature of MMD mainly includes genetic mutation, ring finger protein 213 (RNF213), and dysregulation of the immune response [3,4,5]. However, the etiology and pathophysiological changes in the MMD vessels remain largely unknown.

The typical pathological changes in the MMD stenosis vessels comprise hyperplasia (of the intima) accompanied by the shrinkage of the media layer [6]. Accumulating evidence illustrates that the neointima plays a crucial role in the vessel stenosis of MMD [7]. Previous studies revealed that vascular smooth muscle cells (VSMCs) and endothelial cells (ECs) participate in the neointima of MMD [8,9,10]. Various studies have pointed out that the RNF213 mutation, *4810K*, for instance, affected the angiogenesis of ECs and the proliferating activity of VSMCs, indicating that genetic changes largely influence the general properties of VSMCs or ECs [11,12,13]. Other groups also demonstrated that numerous genes are up-regulated in the MMD vessels, for example, platelet-derived growth factor subunit B(PDGFb) and IL-1beta [14,15].

It has been intensively investigated that noncoding RNA (ncRNA), including circular RNA (circRNA), microRNA (miRNA), and long noncoding RNA (lncRNA), regulate cellular processes via RNA–RNA interaction or RNA–protein interaction [16,17,18,19]. Less is known about the circRNA–miRNA–mRNA regulative network in MMD. Therefore, we aim to establish the circRNA–miRNA–mRNA regulative network in MMD based on RNA sequencing and the “sponging-like” interaction mechanisms [20,21,22] to investigate the potential regulative axis in VSMC- or EC-mediated neointima in MMD.

## 2. Materials and Methods

### 2.1. Clinical Specimen Collection and Ethics Statement

The present study was conducted in accordance with the Declaration of Helsinki and was approved by the Medical Ethics Committee of Tongji Hospital (no.TJ-IRB20220834). Written informed consent was obtained from all the individuals included in the study. The MMD patients were diagnosed according to the World Health Organization criteria. The clinical specimens of blood vessel tissue and blood were taken from 50 patients who were operated on in the neurosurgical emergency department of Tongji Hospital. Blood samples, drawn just before the operation, were collected in K2 EDTA BD Vacutainer tubes and processed within 12 h of collection. The whole blood sample was centrifuged at 60× *g* for 15 min at room temperature, and then the plasma was collected and stored at −80 °C. The expression of miR-125a-3p and circZXDC in the plasma were detected by quantitative, real-time PCR.

### 2.2. Network Establishing

To establish the axis in MMD, circRNA-seq (Zhao et al.) [17], miRNA-seq (Dai et al.) [16], and mRNA-seq (Kanamori et al.) [23] in MMD were collected. The top 10 up-regulated circRNAs and top 20 down-regulated miRNAs were selected, and the interaction between them was analyzed by the online tool CircBank (http://www.circbank.cn/ accessed on 25 April 2019) according to their binding sites. The top 50 up-regulated mRNAs and the potential regulative relationship between the down-regulated miRNA and up-regulated mRNA were analyzed by TargetScan Release 8.0 (http://targetscan.org/ accessed on September 2021). After finding all the circRNAs, miRNAs, and mRNAs that interacted in MMD, Cytoscape (ver. 3.9.0, National Institute of General Medical Sciences, Bethesda, MD, USA) was used to draw the network.

### 2.3. Reverse Transcription-Quantitative PCR (RT-qPCR)

The RNA in the cells and serum was extracted using an RNA extracting solution (G3013; Wuhan Servicebio Technology Co., Ltd., Wuhan, China) according to the manufacturer’s protocol. The miRNA and circRNA were reverse transcribed with the Hifair® III 1st Strand cDNA Synthesis Kit (gDNA digester plus) (11139ES10; Shanghai Yeasen Biotechnology Co., Ltd., Shanghai, China), and the mRNA was reverse transcribed into cDNA using the Hifair® III 1st Strand cDNA Synthesis Supermix (11141ES10; Shanghai Yeasen Biotechnology Co., Ltd. , Shanghai, China) in a 20-µL system. The procedure was as follows: 25 °C for 5 min, 55 °C for 15 min, and 85 °C for 5 min. qPCR was carried out with the Hieff® qPCR SYBR® Green Master mix (11202ES03;11199ESS03; Shanghai Yeasen Biotechnology Co., Ltd., Shanghai, China) on an ABI-Prism 7500 Real-Time PCR System (Applied Biosystems, Carlsbad, CA, USA) and using the following thermocycling conditions: one cycle at 95 °C for 5 min, followed by 40 cycles at 95 °C for 10 s, 60 °C for 20 s, and 72 °C for 20 s. The final melt curve stage at 60 °C for 1 min and 95 °C for 1 s was created using Quant Studio-1™ Design and Analysis Software (A40426; Thermo Fisher Scientific, Waltham, MA, USA). The relative level of miR was standardized with U6(MQP-0201, RiboBio,Guangzhou, China). Other RNAs were normalized to the internal standard, GAPDH. The relative expression level of the DEGs was assessed by the 2^−ΔΔCT^ method: ΔCq = Cq (target gene) − Cq (Reference genes) and ΔΔCq = ΔCq (target gene) − ΔCq (control). The gene-specific primers of the DEGs were designed and commercially synthesized at the Servicebio Technology Company (Wuhan, China). 

### 2.4. Cell Culture

HASMCs and HUVECs were gifted by Dr. Qing Li (Department of Nephrology, Tongji Hospital) and were cultured in Dulbecco’s modified Eagle medium (DMEM; Sigma-Aldrich, Darmstadt, Germany), such as 293T cells, containing 10% fetal bovine serum (FBS; Gibco, NY, USA), and 1% penicillin and streptomycin (Gibco, NY, USA). All cells were grown at 37 °C in a humidified incubator with 5% CO_2_. The HASMCs were randomly divided into: (1) Si-RNF213-1, Si-RNF213-2, Si-RNF213-3, and Si-RNF213-NC; (2) circZXDC overexpression and circZXDC overexpression vector; (3) circZXDC overexpression vector + miR-125a-3p mimic, circZXDC overexpression vector + miR-125a-3p NC, circZXDC overexpression + miR-125a-3p mimic, and circZXDC overexpression + miR-125a-3p NC; (4) Si-ABCC6-1, Si-ABCC6-2, Si-ABCC6-3, and Si-ABCC6-NC, and (5) ABCC6 overexpression, ABCC6 overexpression vector, and ABCC6 overexpression + ERS inhibitor (4-Phenylbutyric acid, 4-PBA, HY-A0281, MCE, NJ, USA) according to the manufacturer’s recommendations with different management.

### 2.5. ROC Curve 

In order to verify if circRNA expression (using ΔCq = Cq (circ0067130) − Cq GAPDH) was a marker for distinguishing MMD patients from control patients, receiver operating characteristic (ROC) curves and the area under the curve (AUC) were calculated using the pROC package, and the circRNA that showed an AUC > 0.75 were considered to be good potential biomarkers. All tests and graphs were performed in R statistical software (ver. 4.1.2., Robert Gentleman, Auckland, New Zealand).

### 2.6. Cell TRANSFECTION

Cells were inoculated in 6-well plates for 24 h prior to transfection, and HASMCs were transfected at 70–80% confluence with (1) si-RNF213 and si-NC (150 nM; Tsingke Biotechnology Co., Ltd., Beijing, China); (2) circZXDC overexpression plasmids and vector plasmids (2 µg and 2 µg, respectively; Shanghai Genechem Co., Ltd., Shanghai, China); (3) miR-125a-3p mimic and mimic negative control (NC) (50 nM and 50 nM, respectively; Tsingke Biotechnology Co., Ltd. , Beijing, China); (4) si-ABCC6 and si-NC (150 nM; Tsingke Biotechnology Co., Ltd. , Beijing, China); (5) ABCC6 overexpression plasmids and ABCC6 overexpression vector plasmids (2 µg and 2 μg, respectively; Tsingke Biotechnology Co., Ltd., Beijing, China), using Lipofectamine™ 3000 reagent (Invitrogen, Carlsbad, CA, USA) according to the manufacturer’s guidance. The HUVEC cells were transfected at 70–80% confluence with si-RNF213 and si-NC. The 293T cells were transfected at 70–80% confluence with si-RNF213 and si-NC. An amount of 5 µL lipo3000 reagent with 125 µL Opti-MEM (Gibco, NY, USA) was added and fully mixed with plasmids or SiRNA, adding 125 µL Opti-MEM at room temperature for 15 min. These were added to the destination cells with a total volume of 1 mL of Opti-MEM. After 24 h incubation at 37 °C, a complete medium change was performed. The effects of gene knockdown or overexpression were confirmed by quantitative real-time PCR or Western blotting. The element of CircZXDC overexpression plasmids is shown as follows: CMV enhancer–left circular frame–MCS–right circular frame–EF1a–ZsGreen1–SV40–puromycin. Coding sequence: GCTGTGAGAAGACATTTATCACAGTGAGTGCCCTGTTTTCCCATAACCGAGCCCACTTCAGGGAACAAGAGCTCTTTTCCTGCTCCTTTCCTGGGTGCAGCAAGCAGTATGATAAAGCCTGTCGGCTGAAAATTCACCTGCGGAGCCATACAGGTGAAAGACCATTTATTTGTGACTCTGACAGCTGTGGCTGGACCTTCACCAGCATGTCCAAACTTCTAAGGCACAGAAGGAAACATGACGATGACCGGAGGTTTACCTGCCCTGTCGAGGGCTGTGGGAAATCATTCACCAGAGCAGAGCATCTGAAAGGCCACAGCATAACCCACCTAGGCACAAAGCCGTTCGAGTGTCCTGTGGAAG. ABCC6 overexpression plasmids: Coding sequence: ATGGCCGCGCCTGCTGAGCCCTGCGCGGGGCAGGGGGTCTGGAACCAGACAGAGCCTGAACCTGCCGCCACCAGCCTGCTGAGCCTGTGCTTCCTGAGAACAGCAGGGGTCTGGGTACCCCCCATGTACCTCTGGGTCCTTGGTCCCATCTACCTCCTCTTCATCCACCACCATGGCCGGGGCTACCTCCGGATGTCCCCACTCTTCAAAGCCAAGATGGTAGCTGCCATCCCTGGGAGCCTGGAACCAGGCAATGTTCGGGGGAGGCAGGGGACAGGCTGGAACCTGGTGAAGTCTTAA. The sequence of miRNA mimics is as follows: miR-125a-3p mimic: F: ACAGGUGAGGUUCUUGGGAGCC; R: CUCCCAAGAACCUCACCUGUUU. The sequences of ABCC6 siRNAs are as follows: si-ABCC6-1: F: GCAACUGGACAGACCUAGATT; R: UCUAGGUCUGUCCAGUUGCTT. si-ABCC6-2: F: GUACAAGUGUGCUGACCGATT; R: UCGGUCAGCACACUUGUACTT. si-ABCC6-3: F: CGAGAGGUCCAUCAAGUCATT; R: UGACUUGAUGGACCUCUCGTT. si-RNF213-1: F: CCGAGAUUGUCUGCAGAAUTT; R: AUUCUGCAGACAAUCUCGGTT. si-RNF213-2: F: GAACAAAUGCUAGAUACGATT; R: UCGUAUCUAGCAUUUGUUCTT. si-RNF213-3: F: CCAAGAAGCACCUAGAUAATT; R: UUAUCUAGGUGCUUCUUGGTT.

### 2.7. Dual-Luciferase Reporter Assay

The miR-125a-3p mimic and mimic NC were transfected with circ0067130 3’UTR-WT, Mut vectors, and *ABCC6* 3’UTR-WT or Mut vectors, into the HEK-293T cells. After 24 h transfection, a dual-luciferase reporter gene assay system (Promega, Madison, WI, USA) was executed to monitor luciferase activity according to the manufacturer’s guidance.

### 2.8. OGD /RP Model

After transfection, the HASMCs were washed with PBS, and the medium was replaced with a serum-free and sugar-free DMEM (G4528, Wuhan Servicebio Technology Co., Ltd., Wuhan, China). Then, the cell plates were placed in a special hypoxia incubator (Sigma–Aldrich, Darmstadt, Germany) with 95% N_2_ and 5% CO_2_ for 6 h at 37 °C. After OGD, the medium was changed to complete medium, and a normal incubator was used to incubate the cells at 37 °C with 5% CO_2_ for 2 h reperfusion for the next management.

### 2.9. Immunohistochemistry (IHC)

The IHC staining of blood vessels was performed using peroxidase elite rabbit IgG kit (Wuhan Boster Biological Technology Ltd., Wuhan, China) to determine the expression and distribution of RES-associated proteins and ABCC6. Briefly, 4 µm-thick sections were dewaxed and rehydrated in a graded alcohol series. The sections selected for antigen retrieval were heated in sodium citrate buffer at pH 9.0 for 15 min at 90 °C. Endogenous peroxidase was quenched with 3% H_2_O_2_; then, the sections were blocked with 5% normal goat serum before incubation overnight at 4 °C with the primary antibody anti-XBP1 (1:200, Proteintech, Wuhan, China), anti-ABCC6 (1:200, Proteintech, Wuhan, China), and anti-P-eif2a (1:200, Proteintech, Wuhan, China). After washing with phosphate-buffered saline (PBS), the sections were incubated for 2 h with the secondary antibody goat anti-rabbit IgG marked by biotinylation. Then, the sections were washed with PBS and incubated for 20 min at 37 °C with the SABC complexes. Finally, the sections were stained with diaminobenzidine (DAB). Nuclei were counterstained blue with Mayer’s hematoxylin (G1004; Wuhan Servicebio Technology Co., Ltd., Wuhan, China). After dehydration, neutral resin was used to seal.

The results of IHC staining were evaluated by ImageJ (ver. 3.9.0, Bethesda, National Institutes of Health, Bethesda, MD, USA). The IHC average optical density score (AOD) was calculated by Integrated density/area. Each component of immunoreactivity was measured on four different representative slide areas (×400 magnification; area equals 0.44 mm^2^ per field) randomly selected using a light microscope (CKX53, Olympus, Tokyo, Japan).

### 2.10. Hematoxylin–Eosin (H&E) Staining

For H&E staining, paraffin embedded sections were dyed with hematoxylin (G1120A, Beijing Solarbio Science and Technology Co., Ltd., Beijing, China) for 5 min after they were deparaffinized and hydrated. Then, they were washed with soft running water, treated with 1% hydrochloric acid alcohol (G1120B, Beijing Solarbio Science and Technology Co., Ltd., Beijing, China) for seconds, and quickly turned blue with 0.6% ammonia. Then, the slices were dyed in the eosin dye solution (G1120C, Beijing Solarbio Science and Technology Co., Ltd., Beijing, China) for 4 min. After dehydration, neutral resin was used to seal, and the images were captured by a light microscope (CKX53, Olympus, Tokyo, Japan).

### 2.11. Immunofluorescence 

The VSMCs were fixed with 4% PFA, and then 0.1% Triton X-100 in PBS was used to permeabilize the cells for 10 min. Nonspecific binding was blocked with 5% normal donkey serum for 2 h. As for the blood vessel paraffin sections, after dewaxing and rehydrating, the microwave antigen repair technique was used. Then, the sections were blocked the same as the cells. The cells were incubated overnight with the primary antibody rabbit anti-ki67 (#9129,1:1000, Cell Signaling Technology, Danvers, MA, USA), and the sections were incubated overnight with the primary antibody rabbit anti-ABCC6 (27848-1-AP, 1:200, Proteintech, Wuhan, China), mouse anti-alpha-actin (sc-32251, 1:200, Santacruz, Dallas, TX, USA), and rabbit anti-xbp1 (24868-1-AP, 1:200, Proteintech, Wuhan, China). The following day, the primary antibody was washed with PBS and the samples were incubated with the secondary anti-rabbit IgG (H + L), F(ab’)2 fragment conjugated to Alexa Fluor 594 fluorescent dye and anti-mouse IgG (H + L), and F(ab’)2 fragment conjugated to Alexa Fluor 488 fluorescent dye (#8889, #4408, 1:1000, Cell Signaling Technology, Danvers, MA, USA) for 1 h for immunofluorescence detection. Then, they were counterstained with DAPI (cat. No. GDP1024; Wuhan Servicebio Technology Co., Ltd., Wuhan, China). The Antifade Mounting Medium (cat. No. G1401; Wuhan Servicebio Technology Co., Ltd., Wuhan, China) was used before being sealed with a cover glass. Images were captured using a fluorescence microscope (CKX53; Olympus Corporation, Tokyo, Japan). Software ImageJ (ver. 3.9.0, Bethesda, National Institutes of Health, Bethesda, MD, USA) was used for data analysis.

### 2.12. Western Blotting

Proteins extracted from the HASMCs of each group were used in Western blotting to validate the RES-related proteins and the transdifferentiation level under different conditions. Total proteins were extracted from the cells with RIPA (G2002; Wuhan Servicebio Technology Co., Ltd., Wuhan, China) lysis buffer (RIPA: PMSF: phosphoproteins inhibitor A:B = 100:1:1:1) on ice. The ultrasonic cell crushing apparatus was used to crush the cells for 1 min. To obtain pure supernatant, the cells were centrifuged at 13,000 rpm/min for 10 min at 4 °C. The BCA method was used to determine the concentration of the proteins via a microplate reader (Infinite F50; Tecan Group Ltd., Myron, Zurich, Switzerland) at a wavelength of 570 nm, and then the proteins were denatured by boiling for 10 min. Equal amounts (20 µg) of total protein were separated by SDS-PAGE on 6%, 10%, and 12% gels and transferred onto Immobilon®-P PVDF membranes (IPVH00010; Millipore Sigma, Darmstadt, Germany). The membranes were blocked with NcmBlot blocking buffer (NCM Biotech, Suzhou, China) and then incubated with the following primary antibodies overnight at 4 °C: anti-ABCC6 (27848-1-AP, 1:1000, Proteintech, Wuhan, China), anti-osteopontin (OPN, 22952-1-AP, 1:1000, Proteintech, Wuhan, China), anti-vimentin (VIM, 10366-1-AP, 1:1000, Proteintech, Wuhan, China), anti-EREG (A16372, 1:1000, ABclonal Biotech Co., Ltd., Shanghai, China), anti-xbp1 ( 24868-1-AP, 1:1000, Proteintech, Wuhan, China), anti-GRP78 ( 11587-1-AP, 1:1000, Proteintech, Wuhan, China), anti-GAPDH (A19056, 1:2000, ABclonal Biotech Co., Ltd., Shanghai, China), anti-β-actin (AC026, 1:2000, ABclonal Biotech Co., Ltd., Shanghai, China), and b-tubulin (GB11017, 1:1000, Wuhan Servicebio Technology Co., Ltd., Wuhan, China). On the following day, the secondary antibodies were incubated (HRP-goat anti rabbit, HRP-goat anti mouse, 1:5000, CST) at room temperature for 1 h. ECL was used for protein imaging and development.

### 2.13. Cell Counting Kit-8 (CCK-8) Assay

The viability of the VSMCs was evaluated using CCK-8 (Cat. No. GK10001, Beyotime, Shanghai, China) in accordance with the manufacturer’s instructions. A total of 5 × 10^3^ cells were seeded on 96-well plates, with four repeating wells in each group of experiments. Subsequently, CCK-8 regent was added to each well at 10 µL/0.1 mL for 2 h incubation at 37 °C in a cell incubator; the 96-well plate was analyzed using a microplate reader (Infinite F50; Tecan Group Ltd., Myron, Zurich, Switzerland) at a wavelength of 450 nm.

### 2.14. Transwell Assay

An amount of 1 × 10^4^/100 μL cells were plated in the upper chambers of a polycarbonate filter (pore size, 8 µm; Corning, NY, USA) in 200 µL serum-free DMEM medium. Culture medium, with 10% FBS, was added into the lower chamber, and both chambers were incubated at 37 °C for 12 h. Subsequently, the upper side of the membrane was fixed with 4% paraformaldehyde for 30 min at room temperature and the top layer of the filter was scrubbed with a sterile cotton swab, and the invading cells on the bottom surface were stained with crystal violet (cat. no. G1014; Wuhan Servicebio Technology Co., Ltd., Wuhan, China) for 30 min at room temperature. The cells were examined, counted, and imaged using digital microscopy. Four random fields of each chamber of cells were counted, and the average number of cells was calculated.

### 2.15. Wound-Healing Assay

An assay was conducted to evaluate the migratory behavior of the VSMCs. Parallel horizontal lines were drawn on the back of the 6-well plate; an appropriate amount of uniformly mixed cells was added to the wells; the cells were 100% confluent after overnight culture. A 200 µL pipette tip was used to draw three lines in the middle of each bottom, perpendicular to the parallel line on the back; they were washed with PBS twice to remove the sloughed cells, and then serum-free medium was added to continue culturing. Pictures were taken at 0 h and 24 h, and the distance of the wounds was documented.

## 3. Result

### 3.1. A CircZXDC–miR-125a-3p–ABCC6 Axis Was Identified in the MMD Vessels, and the Plasma Level of CircZXDC Could Be Used as A Diagnostic Biomarker of MMD

In order to establish the circRNA–miRNA–mRNA axis in MMD, circRNA-seq (Zhao et al.) [17], miRNA-seq (Dai et al.) [16], and mRNA-seq (Kanamori et al.) [23] from MMD was collected, and the interaction between the up-regulated circRNA and down-regulated miRNA was analyzed via the online tool CircBank (http://www.circbank.cn/, accessed on 25 April 2019) (according to their binding sites). Furthermore, the potential regulative relationship between the down-regulated miRNA and up-regulated mRNA was analyzed by TargetScan (http://targetscan.org/, accessed on September 2021). The interaction networks were then screened out, as shown in Figure 1A,B. Interestingly, only two circRNAs, along with three miRNAs, which potentially interact with mRNAs in MMD, were identified (Figure 1B and Table 1). We first detected the expression pattern of the two identified circRNAs (hsa_circ_0067130 and hsa_circ_0004508) in the plasma of non-MMD and MMD patients. A significantly higher level of hsa_circ_0067130 (also named circZXDC) was indeed observed in MMD compared to non-MMD plasma, while no remarkable difference was observed for hsa_circ_4508 (Figure 1C). Hence, we selected the circZXDC–miR-125a-3p–ABCC6 axis as the potential target for MMD. 

CircRNA is relatively stable and can be used as a biomarker with diagnostic value for multiple diseases. We determined whether the circZXDC expression level in plasma could discriminate MMD patients from non-MMD patients. The median level of the circZXDC delta CT value (1.384) detected in the plasma of the MMD patients was set as the cut-off point. As presented in Figure 1D, the circZXDC level efficiently discriminated the MMD patients with an AUC value larger than 0.9. In addition, it has been widely recognized that RNF213 is an important susceptibility gene for MMD, especially among the East Asian population. Therefore, we evaluated the diagnostic value of circZXDC in patients with the RNF213 *4810k* wild type (WT) or RNF213 *4810k* mutation (MUT). Indeed, circZXDC successfully discriminated the MMD patients from the non-MMD patients regardless of the RNF213 mutation (Figure 1D). Furthermore, the expression of circZXDC was also compared between those MMD patients with the RNF213 *4810k* wild type (WT) or RNF213 *4810k* mutation (MUT). Interestingly, a notably higher level of circZXDC was identified in the MMD patients with RNF213 *4810kMUT* (Figure 1E) when compared to the non-MMD patients, as well as the RNF213 WT patients. Additionally, RNF213 WT also exhibited significantly increased circZXDC expression compared to non-MMD (Figure 1E). When taken together, our findings provide evidence that the expression level of circZXDC can sensitively indicate MMD patients, which implies that circZXDC might play a role in the progression of MMD with the RNF213 mutation. 

Since both the VSMCs and ECs are the two major components of neointima in MMD, in order to investigate the source of circZXDC in MMD, we first silenced the RNF213 in the VSMCs, ECs, and HEK293T cells. Surprisingly, after RNF213 silencing, circZXDC was remarkably expressed in the VSMCs but not in the ECs or HEK293T cells (Figure 2A), which suggested that VSMC RNF213 knock-down mimics a similar observation in RNF213 *4810kMUT* patients and possibly implies that VSMCs might be the major contributor to circZXDC up-regulation in MMD. A previous study revealed that VSMCs predominantly consist of a proliferative cellular context in the neointima of MMD [24,25]. In order to further assess their relationship, a correlation was performed between the circZXDC expression level in the plasma and the intima thickness. Indeed, the STA vessels from the MMD patients exhibited a much thicker intima than the non-MMD STA vessels (Figure 2B); their thickness was positively correlated with the circZXDC expression level in the plasma. In addition, we validated that VSMCs are the major cellular context in the hyperplastic intima when using immunofluorescence staining (Figure 2C). When taken together, the up-regulation of circZXDC might play a role in VSMCs, and our next investigation focused on the VSMCs. 

### 3.2. CircZXDC–miR-125a-3p Axis Regulates VSMC Transdifferentiation towards the Synthetic Phenotype

VSMCs are a highly plastic cell type, which can be defined as a distinct phenotype according to their function. In general, VSMCs are classified under a contractile/differentiated state and a synthetic/dedifferentiated state. Unlike the contractile phenotype, synthetic VSMCs exhibit potent, proliferative, migrating activity, as well as extracellular matrix synthesis, which are considered crucial for neointima. In MMD, our previous study observed that the VSMCs in the intima of the MMD vessels predominantly displayed the synthetic state [26]. In order to study the effect of circZXDC on VSMCs, circZXDC was overexpressed (Appendix A) in VSMCs, and oxygen glucose deprivation (OGD) was applied to mimic the microenvironment of the MMD vessels after occlusion. The proliferative activity of VSMCs was assessed by CCK-8 kit and Ki-67 staining. As presented in Figure 3A,B, circZXDC overexpression led to an increased level of VSMC proliferation, both in the normal cells and in the presence of OGD. Moreover, significant, potent, migrating activity was also observed in circZXDC overexpressed VSMCs (Figure 3C,D). Furthermore, we determined the signature proteins of the VSMC synthetic phenotype after circZXDC overexpression. Indeed, CircZXDC overexpression led to a VSMC synthetic phenotype switch; meanwhile, there were even stronger signals in the OGD treatment (Figure 3E). In summary, circZXDC overexpression in VSMCs resulted in a synthetic phenotype switch, and the effect was even stronger under a hypoxic microenvironment. 

Our established network indicated that miR-125a-3p might be a down-stream regulative target of circZXDC. Therefore, first, we observed a significant reduction in the miR-125a-3p induced by circZXDC overexpression (Figure 4A). A luciferase assay was performed to determine the direct binding between circZXDC and miR-125a-3p (Figure 4B). Then, we treated the VSMCs with miR-125a-3p in the presence of circZXDC overexpression. It was found that miR-125a-3p efficiently attenuated circZXDC overexpression-mediated enhanced proliferation (Figure 4C,D) and migration (Figure 4E). Furthermore, the synthetic phenotype switch achieved by circZXDC overexpression was largely alleviated by miR-125a-3p treatment (Figure 4F). When taken together, our data demonstrated that the circZXDC–miR-125a-3p axis affects the VSMC phenotype. 

### 3.3. miR-125a-3p Regulates VSMC Transdifferentiation via ABCC6 mRNA Sponging

Based on our established axis, in order to validate ABCC6 as a potential regulative target of miR-125a-3p, we first examined the ABCC6 expression of the VSMCs after circZXDC overexpression and miR-125a-3p treatment with or without OGD. As expected, circZXDC overexpression caused the up-regulation of ABCC6, and this effect was remarkably attenuated in the presence of miR-125a-3p treatment (Figure 5A). In addition, the luciferase assay was used to validate the direct bindings between miR-125a-3p and ABCC6 mRNA (Figure 5B). 

Furthermore, we evaluated the expression pattern of ABCC6 in the MMD vessels. ABCC6 was expressed more highly in the MMD (compared to the non-MMD) vessels, and, taking advantage of the VSMC-specific marker a-SMA (Figure 5C), we observed strong colocalization between ABCC6 and a-SMA, indicating that ABCC6 was indeed expressed more highly in VSMCs in MMD. In order to demonstrate the role of ABCC6 on VSMCs, ABCC6 was knocked down by siRNA. A proliferation and migration assay illustrated that ABCC6 silencing caused a notable decrease in the corresponding activities (Figure 5D–F). In addition, the expression of synthetic markers was also attenuated by ABCC6 knock-down (Figure 5G). When taken together, miR-125a-3p regulates VSMC transformation towards the synthetic state by sponging ABCC6. 

### 3.4. ABCC6 Affects VSMC Transdifferentiation via Regulating Endoplasmic Reticulum Stress

The ABCC6-encoded protein is a member of the superfamily of ATP-binding cassette (ABC) transporters. Its mutation causes vascular disease, such as pseudoxanthoma elasticum. Previous studies have illustrated the close relationship between ABCC6 and ERS [27,28]. Therefore, we first determined the expression pattern of the relative ERS markers in MMD vessels. As expected, the MMD vessels expressed a more intense ERS marker, including *p*-EIFa and XBP1s (spliced Xbp1; when endoplasmic reticulum stress occurs, X-box binding protein 1(Xbp1) mRNA unconventionally splice to generate Xbp1s) compared to the non-MMD vessels (Figure 6A). In addition, ABCC6 expression was also determined, and in order to verify the relationship between ABCC6 and ERS, a correlation analysis between ABCC6 and the ERS marker XBP1s was performed. As presented in Figure 6B, ABCC6 expression positively correlated with the XBP1 expression in the MMD vessels, and their expression was also positively correlated with the thickness of the intima. These results imply that ABCC6 was correlated with ERS and neointima. 

In order to further verify this, ABCC6 was knocked down in the VSMCs, and we examined the expression of the ERS markers. Similar to previous studies [29,30], OGD induced ERS in the VSMCs, while here, we observed that ABCC6 deficiency significantly attenuated ERS extent in VSMCs (Figure 6C). Moreover, ABCC6 overexpression (Appendix A) induced the VSMC synthetic phenotype switch. This effect was strikingly rescued by ERS inhibition with 4-PBA treatment (Figure 6D–F). When taken together, our data demonstrated that ABCC6 amplified the OGD-induced VSMC phenotype switch toward the synthetic state by enhancing ERS. 

## 4. Discussion

The mechanisms behind intima hyperplasia in MMD remain largely unknown. The main concept regarding the physiopathological process of MMD has been attributed to gene mutation-induced cellular dysregulation [31,32]. Noncoding RNAs participate in a wide range of biological processes and execute their function by directly or indirectly interacting with mRNAs or proteins. Due to their specific binding sites, the circRNA–miRNA–mRNA network is considered to be an important regulative signal. However, there are few studies focusing on this interacting axis in VSMCs, especially in MMD. 

Herein, we performed a meta-analysis based on published circRNA-seq, miRNA-seq, and mRNA-seq datasets regarding MMD (in combination with online tools), with the circZXDC–miR-125a-3p–ABCC6 axis identified [17,23]. CircZXDC is a circular form of the ZXD family zinc finger C (ZXDC) gene [33,34], and its function is still unclear. Current knowledge of circRNA has characterized it as noncoding RNA, and an increasing number of circRNAs have been found to be associated with pretranscription/post-transcription processes by acting as miRNA sponges, protein decoys, splicing regulators, and transcription and chromatin modifiers [35,36,37]. Because of the characteristics of circRNA, including its stable structure due to its covalently closed loops [38], redundant expression level, and easy detection, it is currently considered an optimal diagnostics biomarker for various diseases [39,40,41]. In MMD, very few diagnostic biomarkers have been identified and characterized. Therefore, in this study, we tested and validated the diagnostics value of circZXDC in MMD. Our data proved that circZXDC could be a relatively sensitive and specific biomarker for MMD, which can potentially be used for diagnostics. Our study, for the first time, identified a potential biomarker for MMD; however, due to limited prognosis data, this study was unable to further verify the prognosis value of circZXDC. Interestingly, we observed that patients with an RNF213 mutation tend to express a higher level of circZXDC. RNF213 was revealed as an E3 ligase containing a Zinc finger motif, which several groups have recently unveiled as playing a key role in ubiquitination [42]. Therefore, further study is needed to discover the mechanisms (in detail) that show how RNF213 affects circZXDC expression. 

Since there is no information on circZXDC function, we investigated the role of circZXDC in VSMCs and surprisingly revealed its prosynthetic phenotype effect under OGD conditions. The VSMC synthetic state is recognized as a major reason for vascular remodeling in occlusive vascular diseases, such as atherosclerosis [43]. Synthetic VSMCs display potent proliferation and migration activity as well as vigorous collagen synthesis, which contribute to neointima in vascular remodeling [24,25]. In MMD, based on the characteristics of intima hyperplasia, we demonstrated that VSMCs exhibit a synthetic state in the intima of the MMD vessels. Previous publications have extensively reported factors that affect VSMC phenotype change, such as metabolic reprogramming, epigenetic regulation, etc. [44,45,46,47]. Vascular endothelial growth factor (VEGF) and platelet-derived growth factor beta polypeptide b (PDGFbb) have been reported to be associated with MMD vessels and have even been applied to the generation of an MMD-like animal model [48,49,50]. Our results, for the first time, provide evidence that noncoding RNA circZXDC regulates the VSMC phenotype switch by sponging miR-125a-3p. Several groups have reported that circRNA (for example, Hsa_circ_0031608) modulates the VSMC phenotype in intracranial aneurysms [51]. Furthermore, miRNAs, such as miR-103-3p or miR-128-5p, have been found to be related to VSMC function in vascular calcification or atherosclerosis [52,53]. Utilizing overexpression and rescue assays in combination with luciferase assays, our study depicted the regulative axis of circZXDC and miR-125a-3p in VSMC phenotype change under OGD conditions, which further provides more detailed information for MMD neointima. Therefore, disrupting this axis might be a useful approach for reprogramming the state of VSMCs and alleviating proliferative intima. 

The theory of the circRNA/miRNA sponging mechanism refers to the competitive binding of mRNA, which leads to degradation or translation inhibition [54]. Thus, we further validated ABCC6 as targeting mRNA in MMD. The ATP-binding cassette subfamily C member 6 (ABCC6) is mainly found in the basolateral plasma membrane or endoplasmic reticulum of the kidney and liver. Despite the fact that its structural features have been largely characterized, knowledge with respect to its functions was limited to proton pumping, and not much is known regarding its function in VSMCs [55,56]. Recently, several studies have reported that ABCC6 might be associated with endoplasmic reticulum stress (ERS) [27,28], and ERS was considered a key mediator for VSMC transdifferentiation towards the synthetic state [57,58,59]. Interestingly, previous studies have revealed that RNF213 suppression affects ERS, suggesting an intrinsic connection between RNF213 and ERS [60]. Our results demonstrated that ABCC6 induced ERS in VSMCs, which further resulted in the phenotype switch. The underlying reason may be that ABCC6 contributes to proton release, causing a calcium or other ion gradient between the inner or outer membrane of the ER, thus initiating stress signals in the ER. Therefore, despite the fact that it is currently not applicable, developing an ABCC6-specific inhibitor might be a promising therapeutic option for MMD. In the present study, our findings were merely concluded from an in vitro investigation, thus further animal experiments are needed for validation. 

In conclusion, our study illustrated the regulative role of the circZXDC–miR-125a-3p–ABCC6 axis in VSMCs, which contributes to the neointima of MMD vessels. These results provide insight into the pathophysiological mechanisms of MMD, which might form potential targets for developing a new pharmaceutical treatment. 

## Figures and Tables

**Figure 1 cells-11-03792-f001:**
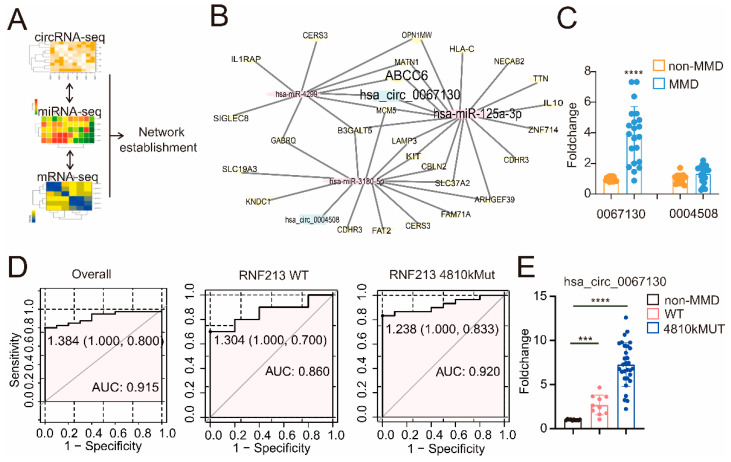
The construction of the circZXDC–miR-125a-3p–ABCC6 axis and the validation of the plasma level of circZXDC as a diagnostic biomarker of MMD. (**A**) A schematic diagram of how the circRNA–miRNA–mRNA network was established. (**B**) Final exact network, including the circRNAs, miRNAs, and mRNAs interacting in MMD. (**C**) The relative expression level of circ0067130 and circ0004508 in the plasma of the non-MMD and MMD patients using RT-qPCR. (**D**) The receiver operating characteristic (ROC) curve analysis for the evaluation of the diagnosis ability of circ0067130 under R. The left panel illustrates the ROC curve of the MMD patients regardless of RNF213 mutation. The middle panel indicates the ROC curve of RNF213 WT MMD patients’ plasma circ0067130. The cut-off point is 1.304, and the AUC value is 0.86. The right panel depicts the RNF213 *p.4810K* mutation in the MMD patient’s plasma circ0067130 ROC curve. The cut-off point is 1.238, and the AUC value is 0.92. (**E**) The fold change of plasma circ0067130 in MMD patients without the RNF213 *p.4810K* mutation and MMD RNF213 *p.4810K* mutation patients compared to non-MMD patients with RT-qPCR. The results of all groups are shown as mean ± SD, and student’s *t*-test was used to compare expression levels or values among different groups, *** *p* < 0.001., **** *p* < 0.0001.

**Figure 2 cells-11-03792-f002:**
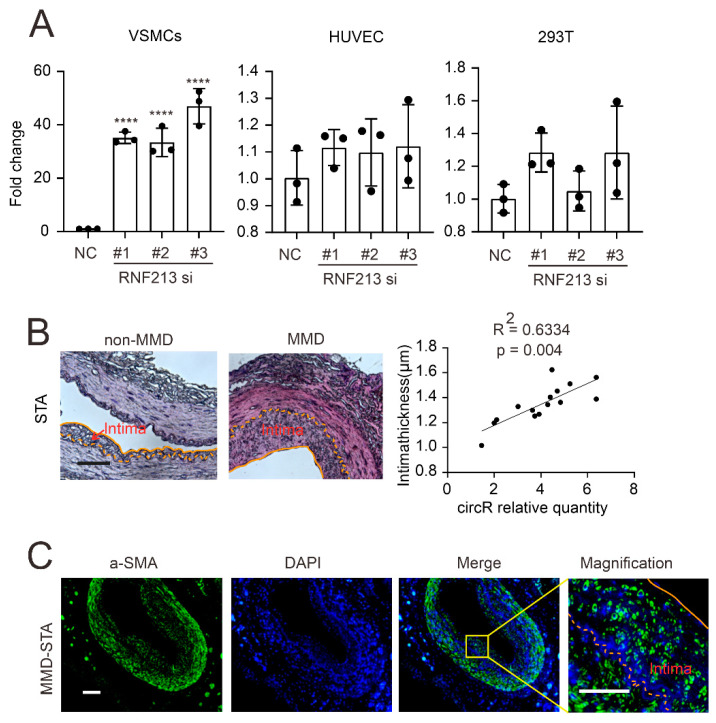
VSMCs contribute to neointima in MMD. (**A**) The RT-qPCR results of the circZXDC relative expression level in VSMCs, HUVEC, and the 293T cells with transferred SiR-RNF213 and NC. (**B**) HE results of STA vessels in non-MMD and MMD patients (intima displayed between the yellow lines). Scale bar = 50 µm. The correlation between circ0067130 expression and intima thickness. R^2^ = 0.6334, and the *p* value is 0.004. (**C**) α-SMA immunofluorescence staining in MMD STA vessels. Scale bar = 100 µm (α-SMA, green; DAPI, blue). **** *p* < 0.0001.

**Figure 3 cells-11-03792-f003:**
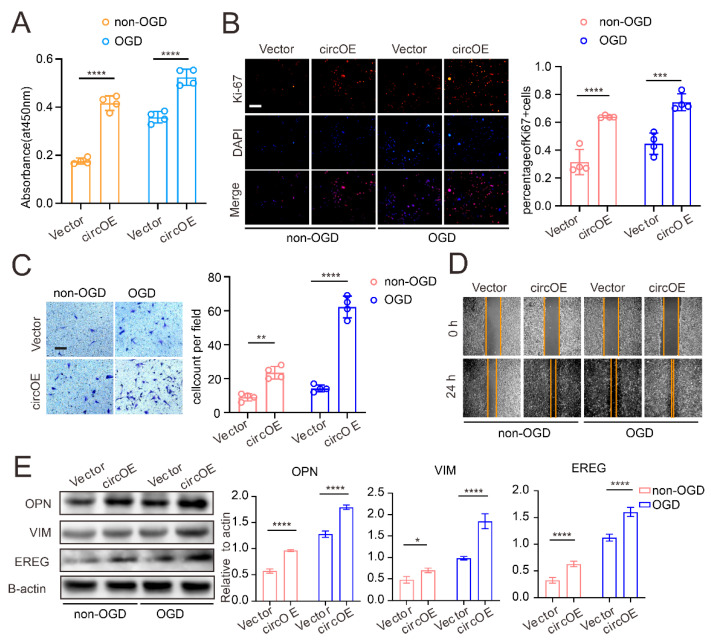
CircZXDC overexpression results in VSMC transdifferentiation towards the synthetic phenotype. The results of the CCK-8 assay (**A**), ki-67 immunofluorescence staining, scale bar = 100 µm (**B**), Transwell assay, scale bar = 50 µm (**C**), and the wound-healing assay (**D**), measuring the VSMCs transferred with the vector plasmid (Vector) and circZXDC overexpression plasmid (circOE) under non-OGD/OGD conditions; (**E**) Western blot analysis of the protein expression of OPN, VIM, and EREG in the VSMCs transferred with the vector plasmid (Vector) and the circZXDC overexpression plasmid (circOE) under non-OGD/OGD conditions. The results of all the groups are shown as mean ± SD; two-way ANOVA with Tukey’s post hoc test was used. The significance level was accepted as * *p* < 0.05, ** *p* < 0.01, *** *p* < 0.001, **** *p* < 0.0001.

**Figure 4 cells-11-03792-f004:**
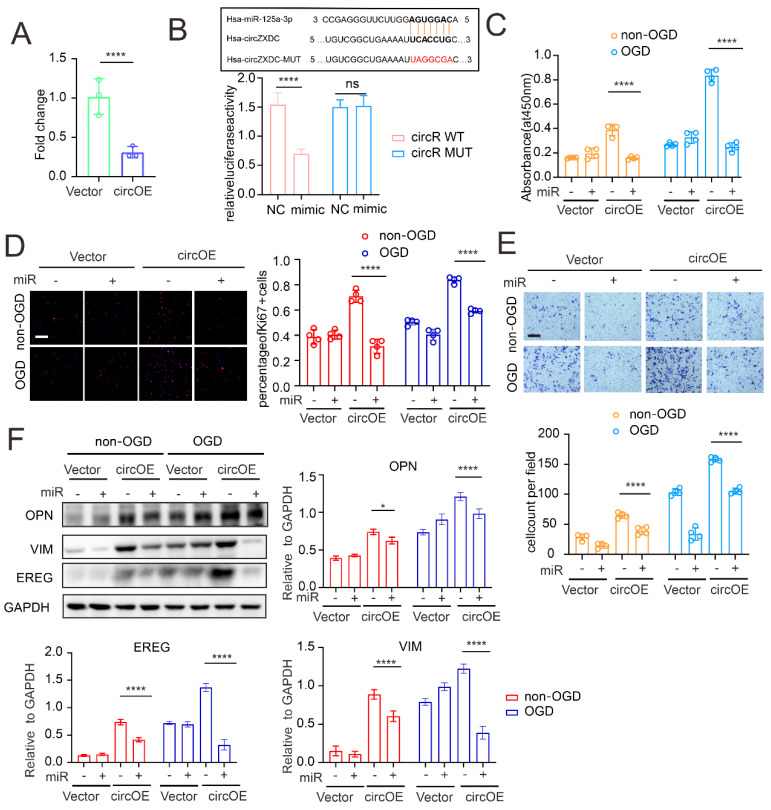
The regulating role of the circZXDC–miR-125a-3p axis in VSMC transdifferentiation towards the synthetic phenotype. (**A**) The RT-qPCR results of miR-125a-3p relative expression in VSMCs transferred with vector plasmid (Vector) and circZXDC overexpression plasmid (circOE). (**B**) The luciferase assay of the targeted relationship between miR-125a-3p and circZXDC. The results of the CCK-8 assay (**C**), ki-67 immunofluorescence staining; scale bar = 100 µm (**D**), Transwell assay; scale bar = 50 µm (**E**); and Western blotting analysis (**F**) of VSMCs transferred with vector plasmid (Vector) or circZXDC overexpression plasmid (circOE) in combination with NC mimic or miR-125a-3p mimic (miR) under non-OGD/OGD conditions. Two-way ANOVA and the Tukey’s post hoc test was used. The significance level was accepted as * *p* < 0.05, **** *p* < 0.0001, ns = no significance.

**Figure 5 cells-11-03792-f005:**
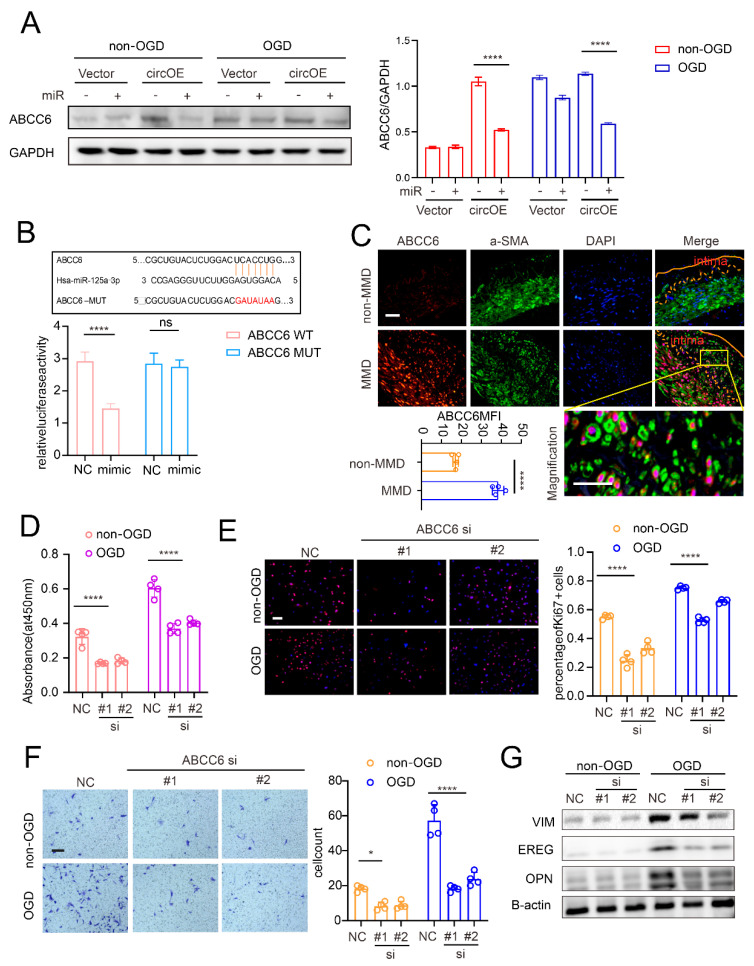
The elucidation of miR-125a-3p regulating VSMCs transdifferentiation via sponging ABCC6 mRNA. (**A**) Western blotting analysis of ABCC6 expression in the VSMCs transferred with the vector plasmid (Vector) or the circZXDC overexpression plasmid (circOE) in combination with the NC mimic or miR-125a-3p mimic (miR) under non-OGD/OGD conditions. (**B**) The luciferase assay of the targeted relationship between miR-125a-3p and ABCC6. (**C**) Immunofluorescence staining of the non-MMD and MMD patient’s STA. Scale bar = 50 µm (ABCC6, red; α-SMA, green; DAPI, blue. Intima circled between the yellow lines). The results of the CCK-8 assay (**D**), ki-67 immunofluorescence staining (**E**), Transwell assay, scale bar = 50 µm (**F**); and Western blotting analysis (**G**) of the VSMCs transferred with NC or ABCC6 siRNA (si) under non-OGD/OGD conditions. The results of all groups are shown as mean ± SD, student’s *t*-test and two-way ANOVA; Tukey’s post hoc test was used. The significance level was accepted as * *p* < 0.05, **** *p* < 0.0001.

**Figure 6 cells-11-03792-f006:**
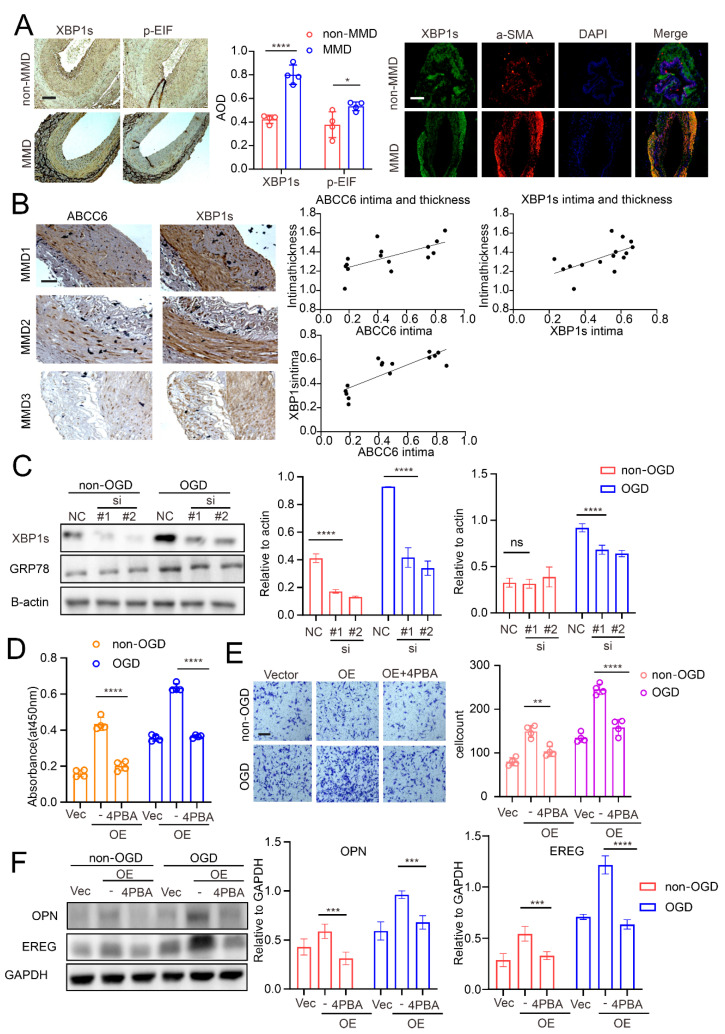
Endoplasmic reticulum stress makes a valuable contribution to the process of VSMC trans-differentiation, affected by ABCC6. (**A**) ERS-related proteins (XBP1s and *p*-EIF), immunohistochemistry, and immunofluorescence staining of non-MMD and MMD patients’ STA vessels, scale bar = 100 µm. (**B**) The correlation between intima ABCC6 expression level and intima thickness (r = 0.665, *p* = 0.0068), the correlation between intima XBP1s expression level and intima thickness (r = 0.611, *p* = 0.0155), and the correlation between ABCC6 expression level and XBP1s expression level in MMD patients’ intima (r = 0.8389, *p* < 0.0001), scale bar = 50 µm. (**C**) XBP1s and GRP78 protein of the VSMCs transferred with NC or ABCC6 siRNA (si) under non-OGD/OGD conditions were detected by Western blotting. The results of the CCK-8 assay (**D**) and Transwell assay (**E**) of the VSMCs transferred with the vector plasmid or the ABCC6 overexpression plasmid (OE) together, with or without 4-PBA, under non-OGD/OGD conditions, scale bar = 50 µm. (**F**) Western blot analysis of the protein expression of OPN and EREG in the VSMCs vector group, ABCC6 overexpression (OE) group, and the ABCC6 overexpression (OE) + 4PBA group under OGD conditions. The results of all groups are shown as mean ± SD. Two-way ANOVA and Tukey’s post hoc test was used. The significance level was accepted as * *p* < 0.05, ** *p* < 0.01, *** *p* < 0.001, **** *p* < 0.0001.

**Table 1 cells-11-03792-t001:** CircRNAs, miRNAs, and mRNAs of the established network based on the MMD RNA-seq datasets.

CircRNA	miRNA	mRNA
hsa_circ_0004508	hsa-miR-4299	ABCC6
hsa_circ_0004508	hsa-miR-4299	SIGLEC8
hsa_circ_0004508	hsa-miR-4299	B3GALT5
hsa_circ_0004508	hsa-miR-4299	GABRQ
hsa_circ_0004508	hsa-miR-4299	OPN1MW
hsa_circ_0004508	hsa-miR-4299	MCM5
hsa_circ_0004508	hsa-miR-4299	CERS3
hsa_circ_0004508	hsa-miR-4299	IL1RAP
hsa_circ_0004508	hsa-miR-4299	MATN1
hsa_circ_0067130	hsa-miR-125a-3p	ABCC6
hsa_circ_0067130	hsa-miR-125a-3p	B3GALT5
hsa_circ_0067130	hsa-miR-125a-3p	NECAB2
hsa_circ_0067130	hsa-miR-125a-3p	MCM5
hsa_circ_0067130	hsa-miR-125a-3p	CBLN2
hsa_circ_0067130	hsa-miR-125a-3p	IL10
hsa_circ_0067130	hsa-miR-125a-3p	HLA-C
hsa_circ_0067130	hsa-miR-125a-3p	ARHGEF39
hsa_circ_0067130	hsa-miR-125a-3p	MATN1
hsa_circ_0067130	hsa-miR-125a-3p	ZNF714
hsa_circ_0067130	hsa-miR-125a-3p	LAMP3
hsa_circ_0067130	hsa-miR-125a-3p	OPN1MW
hsa_circ_0067130	hsa-miR-125a-3p	KIT
hsa_circ_0067130	hsa-miR-125a-3p	SLC37A2
hsa_circ_0067130	hsa-miR-125a-3p	SLC19A3
hsa_circ_0067130	hsa-miR-125a-3p	CDHR3
hsa_circ_0067130	hsa-miR-125a-3p	TTN
hsa_circ_0067130	hsa-miR-3180-5p	KNDC1
hsa_circ_0067130	hsa-miR-3180-5p	B3GALT5
hsa_circ_0067130	hsa-miR-3180-5p	FAM71A
hsa_circ_0067130	hsa-miR-3180-5p	MCM5
hsa_circ_0067130	hsa-miR-3180-5p	CBLN2
hsa_circ_0067130	hsa-miR-3180-5p	FAT2
hsa_circ_0067130	hsa-miR-3180-5p	ARHGEF39
hsa_circ_0067130	hsa-miR-3180-5p	LAMP3
hsa_circ_0067130	hsa-miR-3180-5p	GABRQ
hsa_circ_0067130	hsa-miR-3180-5p	KIT
hsa_circ_0067130	hsa-miR-3180-5p	CERS3
hsa_circ_0067130	hsa-miR-3180-5p	SLC37A2
hsa_circ_0067130	hsa-miR-3180-5p	SLC19A3
hsa_circ_0067130	hsa-miR-3180-5p	CDHR3

## Data Availability

The datasets used and/or analyzed in the current study are available from the corresponding author upon reasonable request.

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
