# Peer review of "CircZXDC Promotes Vascular Smooth Muscle Cell Transdifferentiation via Regulating miRNA-125a-3p/ABCC6 in Moyamoya Disease"

_cells, 2022, doi:10.3390/cells11233792_

Round 1
Reviewer 1 Report
Authors identified circZXDC/miR-125a-3p/ABCC6 axis as a potential pathophysiological mechanism of MMD. They assigned VSMCs to the cells in which the axis plays a role. To validate the supposition, they performed a variety of experiments using both of clinical specimens of MMD patients (although it is slightly regrettable that it is STA instead of a cerebral artery) and human arterial smooth muscle cells. Through the experiments, they disclosed that the axis regulates the differentiation of VSMCs towards synthetic phenotype, which is corresponding to the thickened intima, a well-known character in the artery of MMD patients. Furthermore, the authors suggested that ABCC6, the downstream molecule in the axis, regulates endoplasmic reticulum stress (ERS) by demonstrating an increase in ERS markers.
Although animal studies are required to elucidate whether the axis plays an essential role in the progression of MMD, the authors have demonstrated satisfyingly detailed investigation to show its significance in MMD pathophysiology.
I just propose some minor issues the authors should address to improve the quality of the manuscript.
1. In line 72, it feels like the downstream effectors 125a-3p and ABCC6 appear suddenly. I think it is better to provide a decipherable (larger) figure in panel C in Figure 1. Alternatively, please provide an explanation of how you specified only one axis by fixing circ0067130.
2. The titles of Figures 3 and 4 are the same. It is better to assign titles that describe the contents more specifically, isn’t it?
3. In line 179, 'was also' may be more proper than 'were also'.
4. In figure 7, panel C, please show what the vertical lines mean.
Author Response
Comments and Suggestions for Authors
Authors identified circZXDC/miR-125a-3p/ABCC6 axis as a potential pathophysiological mechanism of MMD. They assigned VSMCs to the cells in which the axis plays a role. To validate the supposition, they performed a variety of experiments using both of clinical specimens of MMD patients (although it is slightly regrettable that it is STA instead of a cerebral artery) and human arterial smooth muscle cells. Through the experiments, they disclosed that the axis regulates the differentiation of VSMCs towards synthetic phenotype, which is corresponding to the thickened intima, a well-known character in the artery of MMD patients. Furthermore, the authors suggested that ABCC6, the downstream molecule in the axis, regulates endoplasmic reticulum stress (ERS) by demonstrating an increase in ERS markers.
Although animal studies are required to elucidate whether the axis plays an essential role in the progression of MMD, the authors have demonstrated satisfyingly detailed investigation to show its significance in MMD pathophysiology.
I just propose some minor issues the authors should address to improve the quality of the manuscript.
- In line 72, it feels like the downstream effectors 125a-3p and ABCC6 appear suddenly. I think it is better to provide a decipherable (larger) figure in panel C in Figure 1. Alternatively, please provide an explanation of how you specified only one axis by fixing circ0067130.
Response: As suggested by the reviewer, we have now put the previous Fig.1B into Supplementary Figure 1and enlarged the Fig.1C for better illustration. Regarding the reason for choosing circZXDC, after network establishment, there are only two circRNAs which were identified to have potential regulative role on miRNAs in MMD. To verify circZXDC, we examined the expression and validate the increase level of circZXDC in MMD plasma compared non MMD patients. we have now added the specific details of network establishment in the method part and listed the established network describing identified miRNAs and mRNAs as well as corresponding circRNA (See Table 1).
- The titles of Figures 3 and 4 are the same. It is better to assign titles that describe the contents more specifically, isn’t it?
Response: Yes indeed. As suggested by the reviewer, we have now revised the titles.
- In line 179, 'was also' may be more proper than 'were also'.
Response: As suggested by the reviewer, we have now revised the expression.
- In figure 7, panel C, please show what the vertical lines mean
Response: As suggested by the reviewer, we have now added the captions.
Reviewer 2 Report
In this study, the authors investigated circRNA/miRNA/mRNA regulative network in moyamoya disease (MMD) using published datasets and online tools. As a result, they identified circZXDC/miR-125a-3p/ABCC6 axis. Plasma levels of circZXDC were elevated in MMD patient, particularly with RNF213 p.R4810K variant. RNF213 suppression upregulated circZXDC specifically in vascular smooth muscle cells (VSMCs), which was associated with neointima in MMD. Cultured VSMC overexpressing circZXDC revealed synthetic/dedifferentiated phenotype. MiR-125a-3p and ABCC6 also play an essential role in the synthetic phenotype switch induced by circZXDC overexpression. The cultured cell model and histopathological analysis showed that ABCC6 induced endoplasmic reticulum stress (ERS), which regulated the synthetic phenotype switch. In conclusion, the authors propose a regulative role of circZXDC/miR-125a- 3p/ABCC6 axis in VSMC, which contribute to neointima of MMD vessels.
This study demonstrated new and interesting findings to provide insights into the pathogenesis of MMD, but I beleave that several revises are required.
Major comment
1. To assess the performance of circZXDC as a diagnostic biomarker for MMD, ROC curve data in the patients with wild-type RNF213 and ROC curve data in the patients with RNF213 R4810K should be shown. Also, the data of non-MMD should be added and compared with RNF213 WT and 4810k MUT in Fig 1F. This is useful to help determine if circZXDC can be adapted to non-Asian MMD patients, most of whom don’t have RNF213 p.R4810K variant.
2. Fig. 1B and 1C are too small and low resolution to understand them and make no sense. Please improve them.
3. The identified miRNAs and mRNAs from your established network should be listed, if they are not confidential.
4. The detailed method of network establishment of circRNA/miRNA/mRNA have to be described in the Method section. Especially, please explain the reason or process to select datasets of Reference 16,17 and 23.
5.A recent study reported that RNF213 suppression inhibits ERS in HeLa cells and mouse embryonic fibroblasts (Biochem Biophys Res Commun. 2022 Jun 18;609:62-68. doi: 10.1016/j.bbrc.2022.04.007.). Therefore, the authors should check the effects of RNF213 suppression on ERS in VSMC or, at least, cite this paper for discussion.
6. It’s better to reduce the part explaining the research background and increase findings of this study in the Abstract.
7. Line 250-252 “Interestingly, our observation that patients with RNF213 mutation tend to express higher level of circZXDC, and further this is also observed in VSMCs.”
This statement is inaccurate because the authors did not express RNF213 p.R4810K but suppressed RNF213 using siRNA in the VSMC experiment. The authors have to delete or revise the sentence.
8. Please show results to confirm the overexpression in VSMC transfected with circZXDC and ABCC6 OE plasmid.
9. Please cite references in the following sentences
Line 105-106 "Previous study revealed that VSMCs predominantly consist of proliferative cellular context in neointima of MMD"
Line 196-197 "Previous studies illustrated the closely relationship between ABCC6 and endoplasmic reticulum stress (ERS)"
Line221 "Similar as previous studies,"
10. What does mean “s” of XBP1s? Spliced form? Please explain in the manuscript.
Minor comments
11. Do “circZXDC expression” at line 107 and 109 mean plasma circZXDC level? If so, this expression is misleading. Please revise them.
12. Please provide more explanation and/or citation of IHC average optical density score (AOD).
13. The authors have to add captions of the pictures of Fig2B.
14. In Fig 4B, DNA sequence of the mimics is too small to read.
15. The authors have to add X-axis labels of the graph of OPN in Fig 4F.
16. Abbreviations should be noted with the full spelling when they appear in the text for the first time. A number of abbreviations do not follows this rule. Please check carefully it.
Author Response
- To assess the performance of circZXDC as a diagnostic biomarker for MMD, ROC curve data in the patients with wild-type RNF213 and ROC curve data in the patients with RNF213 R4810K should be shown. Also, the data of non-MMD should be added and compared with RNF213 WT and 4810k MUT in Fig 1F. This is useful to help determine if circZXDC can be adapted to non-Asian MMD patients, most of whom don’t have RNF213 p. R4810K variant.
Response: As suggested by the reviewer, in addition to overall ROC curve for MMD patients, we have now added the ROC curve data for MMD patients with RNF213WT and RNF213 4810kMut, respectively (See Fig. 1D). Besides, regarding the expression in previous Fig.1F, we have now added the separate expression in RNF213 WT and 4810k MUT patients, as requested by the reviewer (See Fig. 1E). Interestingly, the identified circZXDC displayed diagnostic value for both patients with RNF213WT and RNF213 p. R4810K variant.
- Fig. 1B and 1C are too small and low resolution to understand them and make no sense. Please improve them.
Response: As suggested by the reviewer, we have now put the previous Fig.1B into Supplementary Figure 1and enlarged the Fig.1C for better illustration.
- The identified miRNAs and mRNAs from your established network should be listed, if they are not confidential.
Response: As suggested by the reviewer, we have now listed the established network describing identified miRNAs and mRNAs as well as corresponding circRNA (See Table 1).
- The detailed method of network establishment of circRNA/miRNA/mRNA have to be described in the Method section. Especially, please explain the reason or process to select datasets of Reference 16,17 and 23.
Response: As suggested by the reviewer, we have now added the specific details of network establishment in the method part. Some published RNA-seq data of MMD are unavailable to get further analysis. The reason to choose these datasets is that these are the only MMD RNA-seq datasets available for analysis. Indeed, future study would be a step further to include RNA-seq datasets as many as possible to comprehensively establishing more regulative network that might participate in MMD progression.
5.A recent study reported that RNF213 suppression inhibits ERS in HeLa cells and mouse embryonic fibroblasts (Biochem Biophys Res Commun. 2022 Jun 18;609:62-68. doi: 10.1016/j.bbrc.2022.04.007.). Therefore, the authors should check the effects of RNF213 suppression on ERS in VSMC or, at least, cite this paper for discussion.
Response: As suggested by the reviewer, we have now cited this paper in the discussion part. We thank for the suggestion.
- It’s better to reduce the part explaining the research background and increase findings of this study in the Abstract.
Response: As suggested by the reviewer, we have now revised the Abstract with less research background and more findings. We thank for the suggestion.
- Line 250-252 “Interestingly, our observation that patients with RNF213 mutation tend to express higher level of circZXDC, and further this is also observed in VSMCs.”
This statement is inaccurate because the authors did not express RNF213 p.R4810K but suppressed RNF213 using siRNA in the VSMC experiment. The authors have to delete or revise the sentence.
Response: As suggested by the reviewer, we have now revised this sentence for more accuracy.
- Please show results to confirm the overexpression in VSMC transfected with circZXDC and ABCC6 OE plasmid.
Response: As suggested by the reviewer, we have now added the validation results of overexpression in Supplementary Figure 2.
- Please cite references in the following sentences
Line 105-106 "Previous study revealed that VSMCs predominantly consist of proliferative cellular context in neointima of MMD"
Line 196-197 "Previous studies illustrated the closely relationship between ABCC6 and endoplasmic reticulum stress (ERS)"
Line221 "Similar as previous studies,"
Response: As suggested by the reviewer, we have now added corresponding references. We thank for the suggestion.
- What does mean “s” of XBP1s? Spliced form? Please explain in the manuscript.
Response: XBP1s means spliced XBP1, When endoplasmic reticulum stress occurs, X-box binding protein 1(XBP1) mRNA unconventionally splice to generate XBP1s. As suggested by the reviewer, we have now added the explanation of the “s” in the manuscript.
Minor comments
- Do “circZXDC expression” at line 107 and 109 mean plasma circZXDC level? If so, this expression is misleading. Please revise them.
Response: Yes indeed, they are all mean plasma circZXDC level. We have now revised it in the text.
- Please provide more explanation and/or citation of IHC average optical density score (AOD).
Response: The results of IHC staining were evaluated by software ImageJ (National Institutes of Health, Bethesda, Maryland). IHC average optical density score (AOD) was calculated by Integrated density/Area. We have added it in the method part.
- The authors have to add captions of the pictures of Fig2B.
Response: We have now added the captions.
- In Fig 4B, DNA sequence of the mimics is too small to read.
Response: We have now enlarged the sequence as suggested by the reviewer.
- The authors have to add X-axis labels of the graph of OPN in Fig 4F.
Response: We have now added the captions.
- Abbreviations should be noted with the full spelling when they appear in the text for the first time. A number of abbreviations do not follows this rule. Please check carefully it.
Response: We have now added the full spelling of the abbreviations when appear for the first time.
Round 2
Reviewer 2 Report
I believe the manuscript has been sufficiently improved.
Author Response
Comments: I believe the manuscript has been sufficiently improved.
Response: we thank for the encouragement. We hope that our revision is acceptable.